# Composition of Fatty Acids in Bone Marrow of Red Deer from Various Ecosystems and Different Categories

**DOI:** 10.3390/molecules27082511

**Published:** 2022-04-13

**Authors:** Żaneta Steiner-Bogdaszewska, Katarzyna Tajchman, Piotr Domaradzki, Mariusz Florek

**Affiliations:** 1Institute of Parasitology of the Polish Academy of Sciences, Research Station in Kosewo Górne, 11-700 Mrągowo, Poland; kosewopan@kosewopan.pl; 2Department of Animal Ethology and Wildlife Management, Faculty of Animal Sciences and Bioeconomy, University of Life Sciences in Lublin, Akademicka 13, 20-950 Lublin, Poland; 3Department of Quality Assessment and Processing of Animal Products, University of Life Sciences in Lublin, Akademicka 13, 20-950 Lublin, Poland; mariusz.florek@up.lublin.pl

**Keywords:** *Cervus elaphus*, bone marrow, gas chromatography, fatty acids, living conditions, nutritional status

## Abstract

In this study, the influence of the living conditions of red deer (*Cervus elaphus*) fawns (wild vs. farmed) and effect of the category of free-living animals (fawns vs. does) on the fatty acid (FA) profile of the leg bone marrow was assessed. The composition of FAs in the deer bone marrow was determined by the gas chromatography method. In all groups, oleic acid (18:1 *c*9) was the most abundant in deer bone marrow and comprised of approximately 37% of total FAs. The bone marrow of young wild deer was characterized by a significantly (*p* < 0.001) higher fat content and saturated FAs proportion, while farmed fawns contained more moisture (*p* < 0.005) and fat-free dry matter (*p* < 0.001), as well as more monounsaturated FAs *cis* branched-chain FAs and monounsaturated FAs *trans* (*p* < 0.001). Although no significant (*p* > 0.05) differences were found between fawns, in terms of partial sums of PUFA, a significantly (*p* < 0.001) higher level of the sum of *n*-3 and *n*-6 FAs and more favorable *n*-6/*n*-3 ratio in the bone marrow of wild fawns were determined. In general, the legs of wild fawns were better prepared for wintering than farmed ones. In turn, comparing the category-related FAs composition in the bone marrow of free-living animals, a more favorable profile was observed in the adult (does) than in the young (fawns) animals, as the bone marrow of the wild does was characterized by significantly (*p* < 0.001) lower percentages of saturated FAs and a higher percentage of monounsaturated FAs *cis*.

## 1. Introduction

Bone marrow is a soft, highly blood-supplied spongy tissue present inside medullary cavities in long bones. It is divided into red bone marrow, in which hematopoiesis takes place (the formation of morphotic blood elements), and yellow bone marrow, consisting mainly of adipose cells. The red marrow fills all bones in fetuses and neonates and is present in flat and long bone epiphyses in the postnatal period [1]. The fatty acids of bone marrow lipids have been shown to be used as an energy source and contribute to the formation of new cells [2], especially in young animals [3]. In ruminants, tissue lipids in the prenatal period and young deer contain small amounts of linoleic and linolenic acids [4,5,6]. Therefore, it has been suggested that the youngest animals may be at risk of deficiency of essential fatty acids [7]. It is very important that fawns should be properly nourished, in order to obtain the appropriate body weight before the winter period [8], because their condition determines their survival, not only in the next few months, but also in the following years of adult life [9]. Red deer body weight is a good index of condition for populations living in various habitats. Dzięciołowski et al. [10] revealed that wild red deer fawns, irrespective of sex, had the same body weight up to 6 months of age, and sexual dimorphism, as a significantly heavier body mass, was significantly marked in 1.5 years old males. In older hinds, body weight was stabilized at the age of 2.5 years, while, in stags, it stabilized at the age of 4.5 years. Moreover, in adult hinds participating in breeding, no significant differences in body weight were found with increasing age up to 9.5 years and above.

On the farm, the living and nutritional conditions provided to deer can be modified, unlike in the case of wild animals, which are usually exposed to deficiencies in the food base and low temperatures in winter. It has been shown that deer have a number of adaptations to help them to survive the most demanding periods, e.g., they can slow their metabolism in winter or limit their activity, in order to save energy [11,12,13]. In addition, with the age of the animal, the red marrow transformations in yellow, and its proportion in the bones increases [1]. It has been shown that body fat reserves are initially stored in the bone marrow, next around the kidneys, and, finally, under the skin. Hence, many researchers use the content and composition of fat in the bone marrow as an indicator of the condition and nutritional status of ungulates, due to the fact that it is mobilized as the last energy reserve during starvation [14,15,16,17,18,19,20]. Dzięciołowski et al. [10] observed the decline of stag conditions, both by the decrease of kidney fat index and percentages of males with high deposits of visceral fat during and after the rutting season. In turn, does managed their fat reserves more rationally, which was indicated by the lack of differences in kidney fat index values, visceral fat, and body weight during and after rut. However, frequencies of females with a large amount of visceral fat indicate a positive relationship between the proportions of fatty hinds up to the age of 9.5 years. Similarly to does, the condition of fawns was, in both sexes, maintained at a roughly equal level.

In reindeer, it has been shown that the fatty acid composition in proximal bones (femur) is dominated by saturated fatty acids (SFA), whereas unsaturated fatty acids (UFA), mainly oleic acid (C18:1 *n*-9), predominate in distal bones (metatarsus) [21,22]. The high content of this acid contributes to the low melting point of bone marrow fat, increases its fluidity, thus improving the function of animals’ legs during frosty periods [23,24]. The mechanism of maintaining the softness of fats by increasing their unsaturation degree has been observed in the distal bones of various species of mammals [12,16,17]. Irving et al. [23,25] reported that changes in the bone marrow melting point have been detected in both arctic and tropical mammals. The selective deposition of unsaturated fatty acids in the bone marrow may facilitate their mobilization at low temperatures and in periods of famine. It has been confirmed that adipose tissue lipolysis leads to the preferential release of long-chain polyunsaturated fatty acids (LC PUFA) [26,27].

It should be emphasized that the composition and changes in fatty acids contained in bone marrow fat, in relation to nutritional status, have been insufficiently studied, especially in the case of the youngest animals that are most exposed to adverse climatic conditions. The bone marrow composition profile, apart from, e.g., nutrition [28,29,30,31], climatic conditions [23,24], and physiological state [15,18,19,20], also changes with the age of animals [20,32]. Therefore, it seems interesting to know the fatty acid composition of the bone marrow in younger and older deer.

The aim of the study was to compare the composition of fatty acids in the bone marrow of farmed and wild red deer fawns (effect of the living conditions), as well as wild fawns and 3–6-year-old does (effect of animal category).

## 2. Results

A comparison of the average body weight, mean concentrations of fat, moisture, and fat-free dry matter in the bone marrow (BM) between wild and farmed fawns, as well as between wild fawns and wild does, are shown in Table 1. Although the body mass of the fawns did not differ significantly, a higher value (by 3.64 kg) was recorded for farm deer. The bone marrow of wild fawns was characterized by a higher fat content (by 29.62 pp; *p* < 0.001), in comparison with the BM of farmed fawns, which contained more moisture and fat-free dry matter (by 9.34 and 20.67 pp, respectively; *p* < 0.005). In turn, when comparing the category-related proximate composition of the bone marrow of wild deer (fawns vs. does), a significant difference was noted only for the higher content of fat-free dry matter (by 6.15 pp; *p* < 0.01) in the BM of adults (Table 1).

The applied chromatographic separation allowed for identifying 47 different fatty acids, with the unit share of 35 of them not exceeding 1%. A trace amount of FA C12:0 was detected in the bone marrow of the farmed red deer fawns, and no C17:1 *c*7 was present in the wild fawns (Table 2). The significantly highest level of the total SFAs was determined in the BM of wild fawns, compared to other deer groups (*p* < 0.001). Moreover, the mean percentage of almost SFAs (except C10:0) was significantly higher in the BM of this group (Table 2). The BM fat of wild fawns, compared to farmed fawns, contained significantly (*p* < 0.001) more C14:0, C16:0, C18:0, and C20:0, while significantly (*p* < 0.001) less MUFA *cis*, including C16:1 *c*13, C18:1 *c*9, C18:1 *c*12, and C20:1 *c*13, as well as less fatty acids from the BCFA group (*p* < 0.001) and MUFA *trans* (*p* < 0.001), including C18:1 *t*9 and C18:1 *t*11 (VA) (Table 2 and Table 3). Although no significant differences were found between fawns in partial sums of groups, such as OCFA, PUFA, and TFA, when considering the participation of individual FAs in these groups, the bone marrow of wild fawns contained significantly (*p* < 0.003) more C17:0, ΣC18:2 *trans*, ΣC18:3 *trans*, and such polyunsaturated FAs as C18:2 *n*-6 (LA) and C20:2 *n*-6, as well as almost twice as many C18:3 *n*-3 (ALA) and C18:2 *cis* (Table 2 and Table 3). On the other hand, in the BM of farm fawns, more than two times more CLA (*p* < 0.001) and C22:5 *n*-3 DPA (*p* < 0.005) were shown (Table 3). The observed relationships resulted in a significantly (*p* < 0.001) higher level of the sum of *n*-3 and *n*-6 FAs, as well as a more favorable (lower) *n*-6/*n*-3 ratio in BM of wild fawns (Table 3).

When analyzing the category-related fatty acid profile of wild deer bone marrow (does vs. fawns), a large differentiation was also shown. In such a comparison, significant differences were observed in 21 out of 47 identified FAs. The wild does BM was characterized mainly by a significantly (*p* < 0.001) lower percentage of SFAs (by 12.46 pp), which was a consequence of lower C16:0 and C18:0 proportions (Table 2), as well as a much higher level of MUFA *cis* (by 12.17 pp; *p* < 0.001), especially FAs belonging to this group, such as C16:1 *c*9, C18:1 *c*9, and C18:1 *c*11 (Table 3). Significant differences were also shown in the smaller groups of FAs, such as OCFA and BCFA, because, in the BM of wild fawns, their higher proportions were reported; although, in the total pool of identified FAs, these groups constituted a small percentage, respectively, below 3% and 2% of total FAs (Table 2). Interestingly, no significant differences in total PUFA (including *n*-6 and *n*-3 FAs), MUFA *trans*, and TFA were found between the analyzed groups of free-living deer. Only in individual FAs belonging to these groups were significant differences observed; so, wild fawns BM contained more C20:2 *n*-6 (*p* < 0.033) and C18:1 *t*11 (VA; *p* < 0.011), with less C20:3 *n*-3 (*p* < 0.024), in comparison with the BM of wild does. The high level of SFA in the BM of wild fawns resulted in a significantly (*p* < 0.001) lower proportion of PUFA/SFA (Table 3).

Regardless of the differences shown in the bone marrow of the analyzed groups of red deer (farm fawns vs. wild fawns vs. wild does), the dominant fraction of monounsaturated fatty acids (MUFA *cis*; approximately 60%), followed by saturated fatty acids (SFA; approximately 25%), polyunsaturated fatty acids (PUFA; approximately 4%), and *trans*-fatty acids (TFA; approximately 4%), were determined. Among the individual identified FA groups, the OCFA and BCFA were the minority (2–3%). Oleic acid (C18:1 *c*9) was the most abundant in deer bone marrow and comprised of more than 34% of total FA and around 60% of total bone lipid MUFA *cis*. Within SFAs, palmitic (C16:0) was the most abundant, showing levels of approximately 64%. With regard to PUFA, linoleic (C18:2 *n*-6; averaging 1.49%), conjugated linoleic (CLA; averaging 1.14%), and α-linolenic acid (ALA, 18:3 *n*-3; averaging 0.97%) were the predominant FAs, with percentages of approximately 86% of the total PUFA of deer bone marrow (Table 2 and Table 3).

For a more comprehensive analysis of the results of the animal body weight and bone marrow characteristics, a principal component analysis (PCA) was performed with 18 variables (17 active and 1 additional) and 30 cases. Three principal components (PCs) with eigenvalues exceeding 1 (Kaiser criterion) explained 88.51% of the total variance. PC1, PC2, and PC3 accounted for 45.06%, 33.10%, and 10.35% of the variance, respectively (Table 4).

Figure 1 shows the projection of variables as a two-factor plane (PC1 × PC2). The first component (PC1) exhibits a very high and positive correlation (0.710 < r < 0.810) with eight variables, including fat (%), C16:0, C18:0, SFA, C18:2 *n*-6 LA, C18:3 *n*-3 ALA, *n*-3, and *n*-6 acids, as well as a negative correlation with C18:1 *n*-9 (r = −0.860), MUFA cis (r = −0.740), and FFM (%) (r = −0.732) (Table 5). In turn, the second component (PC2) is positively correlated with C14:0 (r = 0.912) and negatively correlated with PUFA/SFA (r = −0.925), C18:1 *c*11 (r = −0.778), and body weight (r = −0.722). The third component (PC3) exhibits only one negative correlation, i.e., with TFA (r = −0.897).

Additionally, based on the loadings and length of the directional vectors, three apparent groups of parameters can be distinguished (Figure 1). The first group of variables, comprising of SFA, C14:0, C16:0, and C18:0, was distributed in the upper right quadrant (Q1) of the graph, situated in both positive areas of the principal components (PC1 and PC2). The second group included fat (%), C18:3 *n*-3 ALA, *n*-3, *n*-6, C18:2 *n*-6 LA, and PUFAs, which were located in the bottom right quadrant (Q2), as defined by positive PC1 and negative PC2 values. The third group, which comprised of MUFA, PUFA/SFA, C18:1 *c*11, and body weight, was located in the bottom left quadrant (Q3), as defined by negative values of both principal components (PC1 and PC2).

Figure 2 shows the projection of cases, depending on the red deer living conditions and animal category (wild fawns vs. farm fawns vs. wild does) in the coordinate system, as defined by PC1 × PC2. In general, two groups with high spatial clustering can be distinguished. The first group consisted of wild deer fawns’ BM, located in the upper right quadrant (Q1), i.e., designated by positive values of PC1 and PC2, and strongly correlated with saturated fatty acids (Figure 1). The second group consisted of farmed fawns’ BM, located in the upper left quadrant (Q4), which is assigned with negative values of PC1 and positive values of PC2, which characterize the moisture and fat-free dry matter percentages (Figure 1). In contrast, samples from the BM of wild does were the most dispersed and mainly located in both bottom quadrants (Q2 and Q3), which are well-correlated with body weight, monounsaturated, and polyunsaturated FAs (Figure 1).

The analysis of the distribution of the variables in Figure 1 indicates that, due to its close proximity, body weight is strongly positively correlated with C18:1 *c*11, MUFA *cis*, and PUFA/SFA, but negatively correlated with C14:0, C16:0, C18:0, and SFAs (located on the opposite side). The perpendicular position, on the other hand, indicates that there is no association between body mass and, e.g., PUFA, TFA, *n*-6, *n*-3, and other elements. All these relationships are confirmed by the correlation coefficients (Pearson or Spearman) calculated for the analyzed variables, which are presented in Table 6.

## 3. Discussion

The study on red deer has shown a large variation in the content of fat, moisture, and fat-free dry matter in the bone marrow of the animals. The lower concentration of fat and higher moisture content were determined in the farmed deer fawns; although, their body weight was higher than that of the wild young animals. Even so, it may indicate malnutrition, as suggested by Ransom [14], Neiland [33], and Thouzeau et al. [13]. However, a high content of fat-free dry matter in the bone marrow in this group of animals is an indication of a higher level of red, rather than yellow, bone marrow, which explains the high moisture content in this tissue, especially considering the fact that the red fraction predominates in animals at an early stage of development; therefore, they are not fully ready for wintering [3]. In addition, despite the diet supplementation, farmed fawns may have unfavorable proportions of fatty acids in the bone marrow (low degree of unsaturation), which may raise the melting point of the fats and impair their fluidity [19]. Therefore, the legs of farmed fawns could be not sufficiently effective in protecting against the cold.

The highest percentage of oleic acid (C18:1 *c*9) was detected in the bone marrow of the farmed animals. However, the bone marrow of farmed and wild fawns exhibited a much higher proportion of SFAs and lower proportion of MUFAs, compared to wild does. A similar relationship was reported by Soppela and Nieminen [20]. As demonstrated by Christie [34], higher proportions of MUFAs in bone marrow lipids, in comparison with PUFAs, are beneficial for cervids because it contributes to the maintenance of adipocyte fluidity at low temperatures. In addition, the reduced PUFA C18 content in the metatarsal bone marrow of the farmed and wild animals was similar or higher than in malnourished reindeer, which implies high mobilization of these fatty acids from the plant diet, as they are not synthesized in the body [35]. This may be related to the farmed red deer fawns’ prolonged ingestion of doe’s milk, as well as the plant-deficient reindeer diet.

Released PUFAs are probably used to synthesize phospholipids for new cells and may be involved in lymphopoiesis and hematopoiesis into the red parts of the bone marrow [2]. Bone marrow adipocytes may also support the immune system by releasing their PUFAs as precursors of the formation of an extensive membrane and synthesis of eicosanoids [36]. The utilization of limited amounts of PUFAs as an energy source is less probable in ruminants; although, it has been found to intensify during prolonged malnutrition in adult rats [37].

The mean percentages of myristic (C14:0) and pentadecanoic (C15:0) fatty acids in the bone marrow of farmed and wild red deer fawns and does were two-fold higher than that reported by Soppela and Nieminen [20] for reindeer. Over two-fold lower levels of palmitic acid (C16:0), margaric acid (C17:0), and stearic acid (C18:0) were determined in the bone marrow of the animals from all the study groups, in comparison with the values reported by Soppela and Nieminen [20] for reindeer fawns and does. The mean percentage of linoleic acid (C18:2 *n*-6) was lower in the bone marrow of red deer fawns and similar to does’ bone marrow samples, in comparison with the results reported by Soppela and Nieminen [20]. Compared to the results reported by Soppela and Nieminen [20], the percentage of α-linoleic acid (C18:3 *n*-3) was similar in the bone marrow of farmed fawns and two-fold higher in the bone marrow of wild fawns and does. The percentage of arachidonic acid (C20:4 *n*-6) was two-fold higher in the bone marrow of fawns and similar to hinds, in comparison with the results reported for reindeer [20]. The percentages of SFAs in the bone marrow in all deer groups in the present study were substantially lower; however, BCFAs, MUFAs, and PUFAs percentages were considerably higher, in comparison with reindeer. These differences can be due to certain factors, such as the species or age of the animals, as the samples were collected for analyses from 1-month-old reindeer.

The fatty acid composition in the femur bone marrow is dominated by saturated fatty acids (SFAs), whereas MUFAs, especially oleic acid, dominate in the distal parts of the skeleton, e.g., the metatarsus [21,22,36]; this finding was proved in all deer groups in the present study. Sugár and Nagy [36] revealed that, in the metatarsal bone marrow of red deer, the percentages of palmitic and stearic acids decreased to 14% and 4%, respectively, whereas the level of palmitoleic and oleic acids increased to 25% and 56%, respectively. However, the bone marrow of all animals exhibited a similar level of stearic acid. Slightly higher concentrations of palmitic acid were detected in the farmed and wild red deer fawns, and the three animal groups exhibited a similar level of stearic acid. Palmitoleic acid was present in lower amounts in the farmed and wild red deer, while the amount of oleic acid was high, but lower than the values reported by Sugár and Nagy [36].

It has been shown that *trans* FAs in ruminants are produced by the bacterial metabolism of PUFAs in the rumen and are, consequently, present in all fats of these animals, with a maximum content of about 6%. Ruminants produce TFA, which constitutes 4% of the fat in milk [38]. Bacterial desaturation of PUFAs from grasses and vegetables in the rumen produces *trans* double bonds in fatty acid molecules, but with a clear preference for a double bond at position 11 in 18 carbon fatty acids, such as vaccenic acid. Small amounts of *trans* fatty acids from linoleic and linolenic acids, together with *trans* fatty acids with 16, 20, and 22 carbon atoms, are produced in the rumen, as well [39]. The percentage of *trans* fatty acids in the bone marrow of the studied deer ranged between 3.60% and 3.99%.

Conjugated linoleic acid (CLA) isomers have been found in the bone marrow of antelope (1.5%), elk (1.0%), and deer (1.0%) [40]. In the present study, the level of CLA was substantially higher (1.73%) in the bone marrow of farm deer fawns and lower amounts in both groups of wild animals, compared to the aforementioned results. This may be related to the longer period of fawns’ milk feeding by does fed a more nutritive diet on the farm. Cordain et al. [40] also reported a PUFA/SFA ratio ranging from 0.24 to 0.33 in lipids from the metatarsal bone marrow in elk, deer, and antelope. In the present study, the cited range of PUFA/SFA ratio was higher, in comparison with the bone marrow of farm and wild fawns, and similar to the level reported for does. Moreover, the PUFA *n*-6/*n*-3 ratio reported by Cordain et al. [40], between 2.24 and 2.88, was higher than the values found in the present study in the bone marrow of all deer groups.

## 4. Materials and Methods

### 4.1. Experimental Design

The study was conducted on 21 individuals of red deer, including six farmed and nine wild males, in their first year of life (fawns), as well as six non-pregnant wild does aged 3–6 years old. According to the breeding scheme, the investigated animals were aged between 6 and 7 months, as young stags are usually slaughtered, while does are intended for herd reconstruction. The animals were kept at the Research Station of the Institute of Parasitology, Polish Academy of Sciences, Kosewo Górne (Region of Warmia and Mazury; Poland; N: 53°48′; E: 21°23′). The feeding system included a rotational pasture in plots with an area and density recommended by DEFRA [41], FEDFA [42], and Mattiello [43]. Fawns born in a natural way during the grazing period, lasting from April to November in Poland, were included in the study. Initially, the calves were fed with milk by does; later, they ate the vegetation from the pasture. In winter (from December to March), does were fed *ad libitum* with grass haylage or hay with a moderate nutritional value, concentrated feed, based on compressed oats and protein supplements, in the form of rapeseed and soybean concentrates (Eco-pasz, Poland) and Josera Phosphoreimer multi-ingredient licks (Josera, Poland, Appendix A, Appendix A).

The second and third groups of animals comprised of wild individuals living in the area of Strzałowo Forest District, located in the immediate vicinity of the deer farm. The wild red deer individuals were harvested during hunting seasons, in accordance with the applicable Rules of Individual and Population Game Animal Selection in Poland (Polish Journal of Laws, Annex to Resolution no. 57/2005, 22 February 2005).

### 4.2. Sampling

MP 800 sensors, coupled with a Tru-test DR 3000 weight reader (accuracy: ±1%, minimum resolution: 100 g), were used for measurement of the body weight of the farmed animals before slaughter. The weight of the wild red deer was estimated from the carcass weight, which accounts for 67% of the total body weight of red deer [28,44]. The carcass weight was determined in culled and eviscerated animals.

All samples of bones and bone marrow were collected in November 2019. The samples from the farmed animals were collected during routine slaughter, i.e., in the final stage of breeding, after the summer grazing period (between April and October). The wild red deer individuals were harvested during hunting seasons, in accordance with the applicable Rules of Individual and Population Game Animal Selection in Poland (Polish Journal of Laws, Annex to Resolution no. 57/2005, 25 February 2005).

On the day of slaughter, metatarsal bone (*ossa metatarsalia*) was sampled from each animal. The bone was dissected by separating the skin, muscles, and tendons using a stainless-steel knife. Fresh bones were opened using a dental titanium drill, carefully to prevent contamination of the bone marrow, which was collected after removal of the red part and frozen (−80 °C).

### 4.3. Analysis of Fat, Moisture, and Fat-Free Dry Matter in Bone Marrow

A sample of the bone marrow, with an approximate weight of 1.5 g (accuracy up to 1 mg), was transferred into a preweighed cellulose thimble. The percent moisture loss was calculated in accordance with PN-ISO 1442:2000 [45], with the drying method (103°C), using a universal oven Memmert UF30 (Schwabach, Germany). Next, bone marrow fat was extracted with the Soxhlet lipid extraction procedure, with the use of Büchi-B-811 (Flawil, Switzerland) equipment and n-hexane as a solvent [46]. Finally, the percentage of fat was calculated. The cellulose thimble with the remaining fat-free dry matter was dried at a temperature of approx. 103 °C for 1 h and cooled to room temperature in a glass desiccator. Next, the percentage of dry matter was calculated.

### 4.4. Fatty Acid Analysis in Bone Marrow

The profile of fatty acids (FAs) in the bone marrow was determined following fat extraction, with the method developed by Folch et al. [47]. Fatty acid methyl esters (FAMEs) were prepared in the process of transmethylation of 50 mg fat samples with a mixture of concentrated H_2_SO_4_ (95%) and methanol, in accordance with the AOCS Official Method Ce 2-66 [48]. Gas chromatographic (GC) analyses were performed according to Domaradzki et al. [49], using a Varian CG 3900 (Walnut Creek, CA, USA) gas chromatograph, equipped with a flame ionization detector (FID). The FAMEs were separated using a CP 7420 capillary column with 10 0m length, 0.25 mm inner diameter, and 0.25 μm film thickness (Agilent Technologies, Santa Clara, CA, USA). The analysis was performed at increasing temperature values. The following temperature program was used: 50 °C for 1 min, 30 °C/min up to 120 °C, 2 °C/min up to 160 °C, 30 min holding, 1 °C/min up to 200 °C, 5 °C/min up 250 °C, 1 min holding time. The temperatures of the injector and detector were 260 °C and 270 °C, respectively. The carrier gas (hydrogen) flow rate was 2 mL/min, the volume of the injected samples was 1 μL, and the split ratio was 1:50. The FAMEs were identified and quantified based on retention times corresponding to reference mixtures (Supelco 37 Component FAME Mix CRM 47885 Supelco Inc., Bellefonte, PA, USA; CLA methyl ester O5632—Sigma-Aldrich, St. Louis, MO, USA; branched-chain FAME mixture BR2, BR3, BR4—Larodan AB Solna, Sweden). Star GC Workstation Version 5.5. software was used (Varian Inc., Walnut Creek, USA). The fatty acid composition was expressed as a percentage of total FAs. The analyses were performed in duplicate.

The numerical symbols of the identified FAs contained the number of carbon atoms in the FA chain (before the colon), number of double bonds (after the colon), and double bond geometry (*cis* or *trans*). The notation “n” was used for FAs belonging to the *n*-3, *n*-6, and *n*-9 families, where the *n*-number indicates the position of the first double bond, counted from the methyl terminal end of the carbon chain. The ratios and indices of the following groups of fatty acids were calculated: SFA—saturated FAs, even-numbered; OCFA—odd-chain FAs; BCFA—branched-chain FAs; MUFA *cis*—monounsaturated FAs, even-numbered; PUFA—polyunsaturated FAs; *n*-3 and *n*-6 FAs; TFA—*trans* FAs: sum of MUFA *trans*, ∑18:2 *trans*, and ∑C18:3 *trans*; ∑CLA —conjugated linoleic acid: sum of 18:2 *c*9,*t*11 and 18:2 *t*9,*c*11; ∑C18:2 *trans*—sum of non-conjugated 18:2 *t,c/ c,t/ t,t* isomers; ∑C18:3 *trans*—sum of 18:3 *trans* isomers with an unknown position of double bonds, as well as the PUFA/SFA and *n*-6/*n*-3 ratios.

### 4.5. Statistical Analysis

The statistical analyses were performed in Statistica ver. 13 (TIBCO Software Inc., Palo Alto, CA, USA). The statistical differences in the individual variables between two groups of red deer (farmed fawns vs. wild fawns or wild fawns vs. wild does) were analyzed depending on the results of the Shapiro-Wilk test of normality. Normally distributed variables were tested using the unpaired Student’s *t*-test (for two independent groups); otherwise, the Kolmogorov-Smirnov test was applied. The results were expressed as the mean value and standard deviation of the variables. The principal component analysis (PCA) was further applied to visualize data and demonstrate the relationships between variables characterizing the fatty acid profile and composition of the bone marrow of different red deer groups. Moreover, depending on the nature of the variable distribution (normal or non-normal), the relationships between the body mass and bone marrow characteristics were assessed using the Pearson correlation coefficient (r) or Spearman rank correlation coefficient (rS), respectively. All relationships were evaluated at a significance level at *p* < 0.05.

## 5. Conclusions

The diet used on the farm could impact the fatty acid composition in young deer bone marrow. However, it does not contribute to a more favorable composition of fatty acids in the bone marrow of fawns’ legs; therefore, these animals seem less well-prepared for the first winter of their lives than wild fawns. The fatty acid composition of wild red deer bone marrow was more favorable in the adult animals than in the 6-month-old fawns. Therefore, they are less efficiently protected from the cold than adult deer. Additionally, the increase in the body weight of the animals increased the levels of unsaturated fatty acids, at the expense of saturated fatty acids.

## Figures and Tables

**Figure 1 molecules-27-02511-f001:**
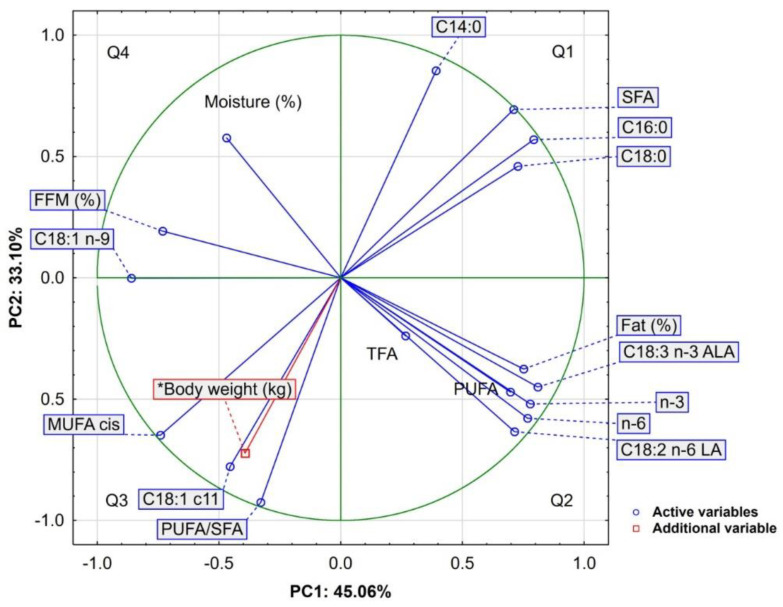
Projection of variables in a two-factor plane (PC1 × PC2); FFM (%)—fat-free mass content; fat (%)—fat content; TFA—*trans* fatty acids.

**Figure 2 molecules-27-02511-f002:**
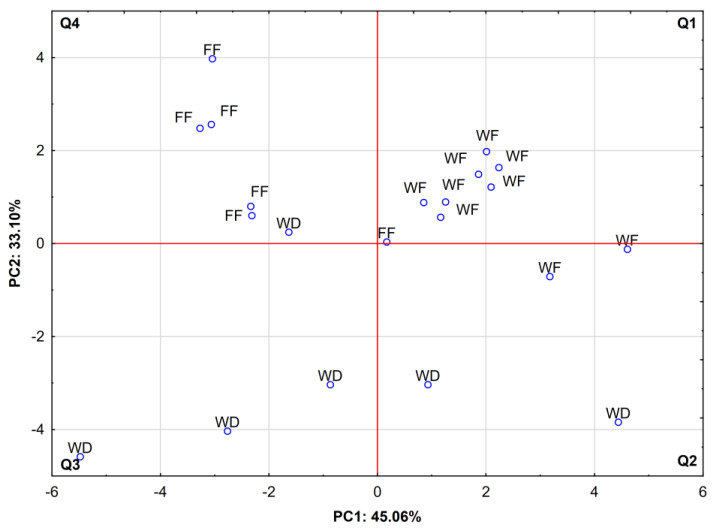
Projection of cases, depending on red deer group, in a two-factor plane (PC1 × PC2); FF—farm fawns; WF—wild fawns; WD—wild does.

**Table 1 molecules-27-02511-t001:** Comparison of the mean body weight, bone marrow composition between red deer fawns and does.

Variable (%)	Farmed Fawns		Wild Fawns		Wild Does		Fawns Wild vs. Farmed *p*-Value t ^a^/K-S ^b^	Wild Fawns vs. Does *p*-Value t ^a^/K-S ^b^
	M	SD	M	SD	M	SD		
**Body Mass (kg)**	49.42	5.12	45.78	6.79	80.83	30.53	>0.05 ^a^	<0.001 ^b^
**Bone Marrow Composition (%)**								
**Fat**	53.52	13.71	83.14	3.43	77.33	9.19	<0.001 ^b^	>0.05 ^b^
**Moisture**	21.01	10.07	11.66	1.77	11.01	4.07	<0.005 ^b^	>0.05 ^b^
**Fat-Free Dry Matter**	25.48	11.78	4.81	2.53	10.96	7.19	<0.001 ^b^	<0.01 ^b^

M—mean, SD—standard deviation, ^a^—Student’s t-test, and ^b^—Kolmogorov–Smirnov test.

**Table 2 molecules-27-02511-t002:** Saturated, odd-, and branched-chain fatty acid composition in the bone marrow of red deer fawns and does.

Variable (%)	Farmed Fawns		Wild Fawns		Wild Does		Wild Fawns vs. Farmed Fawns *p*-Value t ^a^/K-S ^b^	Wild Fawns vs. Wild Does *p*-Value t ^a^/K-S ^b^
	M	SD	M	SD	M	SD		
C10:0	0.06	0.02	0.07	0.02	0.08	0.07	>0.05 ^a^	>0.05 ^b^
C12:0	tr	-	0.62	0.14	0.11	0.05	–	<0.001 ^a^
C14:0	5.18	1.08	5.94	0.64	2.23	0.82	<0.018 ^a^	<0.001 ^a^
C16:0	15.87	0.78	19.39	0.89	13.62	3.57	<0.001 ^a^	<0.001 ^b^
C18:0	4.58	0.73	5.58	0.94	3.85	1.08	<0.003 ^a^	<0.001 ^a^
C20:0	0.12	0.03	0.17	0.03	0.13	0.04	<0.001 ^a^	<0.008 ^a^
ƩSFA	26.27	1.73	31.76	1.90	19.93	5.39	<0.001 ^a^	<0.001 ^b^
C15:0	0.90	0.08	0.89	0.08	0.53	0.16	>0.05 ^a^	<0.001 ^b^
C15:1	0.46	0.05	0.37	0.09	0.36	0.09	<0.05 ^a^	>0.05 ^a^
C17:0	0.49	0.07	0.57	0.06	0.33	0.08	<0.003 ^a^	<0.001 ^a^
C17:1 *c*7	0.09	0.03	-	-	-	-	-	-
C17:1 *c*9	0.73	0.08	0.77	0.07	0.77	0.07	>0.05 ^a^	>0.05 ^a^
C21:0	0.09	0.02	0.06	0.02	0.11	0.04	>0.05 ^a^	<0.001 ^a^
OCFA	2.79	0.23	2.65	0.23	2.07	0.36	>0.05 ^a^	<0.001 ^a^
C13:0 *iso*	0.13	0.05	0.07	0.01	tr	-	<0.001 ^b^	>0.05 ^a^
C13:0 *anteiso*	0.12	0.03	0.11	0.02	tr	-	>0.05 ^a^	>0.05 ^a^
C14:0 *iso*	0.14	0.02	0.07	0.02	0.06	0.02	<0.001 ^a^	>0.05 ^a^
C15:0 *iso*	0.33	0.06	0.18	0.03	0.16	0.04	<0.001 ^b^	>0.05 ^b^
C15:0 *anteiso*	0.52	0.07	0.33	0.07	0.28	0.11	<0.001^a^	>0.05 ^a^
C17:0 *iso*	0.37	0.05	0.27	0.03	0.23	0.03	<0.001^b^	<0.001 ^a^
C17:0 *anteiso*	0.49	0.05	0.48	0.07	0.43	0.08	>0.05 ^a^	>0.05 ^a^
C18:0 *iso*	0.12	0.03	0.14	0.03	0.19	0.05	>0.05 ^a^	<0.025 ^b^
ƩBCFA	2.24	0.29	1.65	0.22	1.39	0.25	<0.001 ^a^	<0.004 ^a^

M—mean, SD- standard deviation, tr—trace: values for peaks < 0.05%, ^a^ —Student’s t-test, and ^b^—Kolmogorov–Smirnov test.

**Table 3 molecules-27-02511-t003:** Unsaturated and *trans* fatty acid composition in the bone marrow of red deer fawns and does.

Variable (%)	Farmed Fawns		Wild Fawns		Wild Does		Wild Fawns vs. Farmed Fawns *p*-Valuet ^a^/K-S ^b^	Wild Fawns Vs. Wild Does *p*-Value t ^a^/K-S ^b^
	M	SD	M	SD	M	SD		
C14:1 *c*9	3.62	0.86	3.15	0.46	2.67	0.95	>0.05 ^a^	>0.05 ^b^
C16:1 *c*11	0.06	0.01	0.06	0.01	0.05	0.01	>0.05 ^a^	>0.05 ^a^
C16:1 *c*13	0.28	0.05	0.18	0.03	0.14	0.04	<0.001 ^a^	<0.008 ^a^
C16:1 *c*7	0.63	0.08	0.57	0.05	0.51	0.03	>0.05 ^b^	<0.001 ^a^
C16:1 *c*9	12.09	2.22	11.35	1.48	14.96	1.71	>0.05 ^a^	<0.001 ^a^
C18:1 *c*9	38.81	1.15	34.98	1.21	37.51	2.48	<0.001 ^a^	<0.005 ^b^
C18:1 *c*11	4.19	0.49	4.46	0.88	10.33	5.66	>0.05 ^b^	<0.001 ^b^
C18:1 *c*12	0.09	0.01	0.14	0.03	0.15	0.04	<0.001 ^a^	>0.05 ^a^
C18:1 *c*13	0.37	0.05	0.42	0.09	0.89	0.42	>0.05 ^a^	<0.005 ^b^
C20:1 *c*9	0.18	0.03	0.24	0.05	0.19	0.05	>0.05 ^a^	>0.05 ^a^
C20:1 *c*11	0.45	0.09	0.46	0.11	0.74	0.32	>0.05 ^a^	>0.05 ^b^
C20:1 *c*13	0.16	0.02	0.12	0.03	0.12	0.02	<0.001 ^a^	>0.05 ^a^
ƩMUFA *cis*	60.91	1.92	56.08	2.29	68.25	6.99	<0.001 ^a^	<0.001 ^b^
C18:2 *n*-6 LA	1.21	0.22	1.59	0.18	1.66	0.32	<0.001 ^a^	>0.05 ^a^
C18:2 *cis*	0.19	0.04	0.36	0.09	0.42	0.04	<0.001 ^b^	>0.05 ^a^
C18:3 *n*-3 ALA	0.62	0.12	1.15	0.26	1.14	0.44	<0.001 ^a^	>0.05 ^a^
CLA	1.73	0.29	0.83	0.21	0.87	0.28	<0.001 ^a^	>0.05 ^a^
C20:2 *n*-6	0.09	0.02	0.13	0.02	0.09	0.03	<0.022 ^a^	<0.033 ^a^
C20:4 *n*-6 AA	0.09	0.04	0.09	0.04	0.07	0.02	>0.05 ^a^	>0.05 ^a^
C20:3 *n*-3	0.07	0.01	0.12	0.03	0.15	0.05	<0.006 ^a^	<0.024 ^a^
C22:5 *n*-3 DPA	0.19	0.10	0.06	0.04	0.06	0.01	<0.005 ^b^	>0.05 ^a^
ƩPUFA	3.94	0.55	4.26	0.46	4.37	0.89	>0.05 ^a^	>0.05 ^b^
C18:1 *t*6/7	0.20	0.02	0.19	0.02	0.17	0.04	>0.05 ^a^	>0.05 ^b^
C18:1 *t*9	0.40	0.07	0.21	0.03	0.21	0.02	<0.001 ^b^	>0.05 ^a^
C18:1 *t*10	0.16	0.02	0.17	0.03	0.21	0.07	>0.05 ^a^	>0.05 ^b^
C18:1 *t*11 VA	1.51	0.30	0.84	0.18	0.65	0.22	<0.001 ^a^	<0.011 ^a^
C18:1 *t*16	0.18	0.02	0.34	0.07	0.32	0.09	<0.001 ^b^	>0.05 ^a^
ƩMUFA *trans*	2.45	0.38	1.75	0.22	1.56	0.39	<0.001 ^a^	>0.05 ^b^
ƩC18:2 *trans*	1.05	0.20	1.39	0.24	1.85	0.37	<0.001 ^a^	<0.001 ^a^
ƩC18:3 *trans*	0.36	0.05	0.45	0.09	0.58	0.03	<0.007 ^a^	<0.001 ^a^
ƩTFA	3.86	0.49	3.60	0.39	3.99	0.68	>0.05 ^a^	>0.05 ^b^
PUFA/SFA	0.15	0.03	0.13	0.02	0.23	0.05	>0.05 ^b^	<0.001 ^b^
*n*-3	0.71	0.20	1.32	0.30	1.33	0.48	<0.001 ^a^	>0.05 ^a^
*n*-6	1.32	0.25	1.74	0.17	1.76	0.34	<0.001 ^a^	>0.05 ^a^
*n*-6/*n*-3	1.95	0.45	1.37	0.29	1.39	0.25	<0.001 ^a^	>0.05 ^a^

M—mean, SD—standard deviation, tr—trace: values for peaks < 0.05%, ^a^ —Student’s t-test, and ^b^—Kolmogorov–Smirnov test, and C18:2 *cis*—this peak may include several 18:2 *cis* isomers with an unknown position of double bonds.

**Table 4 molecules-27-02511-t004:** Eigenvalues and the proportion of variation (%), as explained by the principal components.

Component	Eigenvalue	Proportion	Cumulative
1	7.66	45.06	45.06
2	5.63	33.10	78.16
3	1.76	10.35	88.52
4	0.66	3.85	92.37
5	0.39	2.32	94.69
6	0.30	1.74	96.44
7	0.24	1.39	97.83
8	0.15	0.91	98.73
9	0.10	0.61	99.73
10	0.07	0.39	99.89
11	0.03	0.15	99.94
12	0.01	0.05	99.26
13	0.00	0.03	99.97
14	0.00	0.02	99.98
15	0.00	0.01	99.99
16	0.00	0.00	99.99
17	0.00	0.00	100.00

**Table 5 molecules-27-02511-t005:** Correlations between the principal components and original variables.

Variable	PC1	PC2	PC3
C14:0	0.392	0.854	0.129
C16:0	0.793	0.570	−0.019
C18:0	0.727	0.459	−0.231
C18:1 *n*-9	−0.860	−0.002	−0.227
C18:1 *c*11	−0.454	−0.778	0.205
C18:2 *n*-6 LA	0.714	−0.635	0.002
C18:3 *n*-3 ALA	0.810	−0.450	−0.100
SFA	0.710	0.694	−0.004
MUFA *cis*	−0.740	−0.648	0.145
PUFA	0.698	−0.470	−0.442
TFA	0.266	−0.239	−0.897
*n*-3	0.779	−0.520	−0.097
*n*-6	0.768	−0.579	−0.019
PUFA/SFA	−0.329	−0.925	−0.105
Fat (%)	0.752	−0.376	0.474
Moisture (%)	−0.469	0.577	−0.137
FFM (%)	−0.732	0.192	−0.549
Body weight (kg)	−0.393	−0.722	0.011

**Table 6 molecules-27-02511-t006:** Correlations between the body weight and bone marrow characteristics.

Variable	Correlation Coefficient (^r/rS^)	Significance
C14:0	−0.698 ^r^	***
C16:0	−0.759 ^r^	***
C18:0	−0.581 ^r^	***
C18:1 *n*-9	0.297 ^r^	ns
C18:1 *c*11	0.601 ^rS^	*
C18:2 *n*-6 LA	0.129 ^r^	ns
C18:3 *n*-3 ALA	0.081 ^r^	ns
SFA	−0.767 ^r^	***
MUFA *cis*	0.762 ^r^	***
PUFA	0.011 ^r^	ns
TFA	0.083 ^r^	ns
*n*-3	0.140 ^r^	ns
*n*-6	0.046 ^r^	ns
PUFA/SFA	0.400 ^rS^	*
Fat (%)	−0.122 ^rS^	ns
Moisture (%)	−0.39 ^rS^	ns
FFM (%)	0.284 ^rS^	ns

Correlation coefficient: r—Pearson, rS—Spearman; * *p* < 0.05; ** *p* < 0.01; *** *p* < 0.001; ns—not significant.

## Data Availability

The data presented in this study are available in the article and Supplementary Material.

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
