# Peer review of "Composition of Fatty Acids in Bone Marrow of Red Deer from Various Ecosystems and Different Categories"

_molecules, 2022, doi:10.3390/molecules27082511_

Round 1

Reviewer 1 Report

The topic of the manuscript is original and interesting. The study identified 47 fatty acids in bone marrow of farmed and free-living red deer.

The low numbers (n=6) of samples in two groups and using Anova for the statistical analysis of the data are the most questionable issues. I guess that this was why the authors applied a lot of different tests. The most appropriate variant of the analysis should be chosen. Incorrect description of the statistical analysis in Materials and Methods section leads to many uncertainties. It is unclear what model and what tests were really used. I would like to suggest to refuse of the present patchwork of various tests.

The authors could add new samples in small groups or apply a non-parametric analysis. Moreover, PCA could be combined with both non-parametric and Anova analysis to reveal the differences between the groups particularly for small samples.

Table 1 is very strange and large (displaced in three pages). I guess this is due to an inadequate statistical analysis. As all the sections of the manuscript should be compatible with each other, it is unclear why there is K-S test (not described in methods section) among other unnamed tests. The authors should describe only those methods and tests which were used. Even five columns would be unnecessary if the authors used superscripts to mark the differences between the groups. For the proximate composition and fatty acid composition of bone marrow at least two separate tables could be designed in portrait pages and should be inserted into the main text close to their first citation.

Table 4.: It is unclear why the correlations between the body weight and some variables are presented as Pearson and other variables are calculated as Spearman correlations. Pearson and Spearman correlations could be presented in separate columns.

The presentation of the obtained results in the text should be more consistent in describing groups of animals as well as separate groups of studied variables.

The conclusions can be drawn on the basis of the results obtained. The authors did not study the milking period of either farmed or free-living red deer and did not show age differences of farmed and wild fawns in Methods section, therefore, the conclusions on the impact of longer milking period of farmed deer are not appropriate.

The manuscript in its present status is not suitable for publication. It must be revised.

Author Response

Rec 1

The topic of the manuscript is original and interesting. The study identified 47 fatty acids in bone marrow of farmed and free-living red deer.

Answ. Thank you very much for kind statement.

The low numbers (n=6) of samples in two groups and using Anova for the statistical analysis of the data are the most questionable issues.

Answ. At least six animals from each research group, a total of 21 individuals, were used for the analyzes. In addition, a number of studies have already been prepared in which only six animals were tested, examples below: DOI: 10.1016/j.meatsci.2016.08.008, DOI 10.1007/s11694-017-9479-4, https://doi.org/10.1139/cjas-2018-0234. Moreover it is recommended to carry out research on as few animals as possible.

I guess that this was why the authors applied a lot of different tests. The most appropriate variant of the analysis should be chosen. Incorrect description of the statistical analysis in Materials and Methods section leads to many uncertainties. It is unclear what model and what tests were really used. I would like to suggest to refuse of the present patchwork of various tests.

The authors could add new samples in small groups or apply a non-parametric analysis. Moreover, PCA could be combined with both non-parametric and Anova analysis to reveal the differences between the groups particularly for small samples.

Answ. Indeed, the original description of the statistical analysis was vague and inconsistent, which of course does not excuse the authors. Therefore, we would like to inform that Statistical analysis section was redrafted and clarifying information was included. We thus hope that the description of Table 1 is sufficiently clear to readers and accurately states which variables had a normal distribution (Student's t-test) and for which such a distribution was not found (Kolmogorov-Smirnov test).

“The statistical analyses were performed in Statistica ver. 13 (TIBCO Software Inc., Pa-lo Alto, CA, USA). The statistical differences in the individual variables between two groups of red deer (farmed fawns vs. wild fawns or wild fawns vs. wild does), were analyzed depending on the results of the Shapiro-Wilk test of normality. Variables normally distributed were tested using the unpaired Student’s t-test (for two independent groups), otherwise the Kolmogorov-Smirnov test was applied. The results were expressed as the mean value and standard deviation of the variables. The principal component analysis (PCA) was further applied to visualize data and demonstrate the relationships between variables characterizing the fatty acid profile and composition in the bone marrow and the fallow deer diet. Moreover, depending on the nature of the variable distribution (normal or non-normal) relationships between the body mass and bone marrow characteristics were assessed using the Pearson correlation coefficient (r) or the Spearman rank correlation coefficient (rS), respectively. All relationships were evaluated at a significance level at p < 0.05. An attempt was also made to determine the correlations between the body weight of the animals and the bone marrow characteristics using principal component analysis (PCA).”

Table 1 is very strange and large (displaced in three pages). I guess this is due to an inadequate statistical analysis. As all the sections of the manuscript should be compatible with each other, it is unclear why there is K-S test (not described in methods section) among other unnamed tests. The authors should describe only those methods and tests which were used. Even five columns would be unnecessary if the authors used superscripts to mark the differences between the groups. For the proximate composition and fatty acid composition of bone marrow at least two separate tables could be designed in portrait pages and should be inserted into the main text close to their first citation.

Answ. Table 1 has been improved in line with the recommendations and has been separated into Tables 1, 2 and 3. We hope that the results presented in this way will be clearer.

In addition to the explanations given above concerning the section Statistical analysis we hope that the description of Table 1, 2 and 3 is sufficiently clear to readers and accurately states which variables had a normal distribution (Student's t-test) and for which such a distribution was not found (Kolmogorov-Smirnov test).

Table 4.: It is unclear why the correlations between the body weight and some variables are presented as Pearson and other variables are calculated as Spearman correlations. Pearson and Spearman correlations could be presented in separate columns.

Answ. We would like to inform, that depending on the nature of the variable distribution (normal or non-normal) relationships between the body mass and bone marrow characteristics were assessed using the Pearson correlation coefficient (r) or the Spearman rank correlation co-efficient (rS), respectively.

The presentation of the obtained results in the text should be more consistent in describing groups of animals as well as separate groups of studied variables.

The conclusions can be drawn on the basis of the results obtained. The authors did not study the milking period of either farmed or free-living red deer and did not show age differences of farmed and wild fawns in Methods section, therefore, the conclusions on the impact of longer milking period of farmed deer are not appropriate.

The manuscript in its present status is not suitable for publication. It must be revised.

Answ. Thank you for the advices. The obtained results and conclusions were corrected and rewritten to be more coherent and clearer. The paper has been subjected to linguistic corrections.

Reviewer 2 Report

The manuscript "Composition of fatty acids in bone marrow of red deer from 2 various ecosystems" is well written and thoroughly studied and investigated. Although the novelty of the manuscript is not high, the results of this manuscript in studying the composition of fatty acids in bone marrow of farmed vs. wild red deer fawns vs. 3-6-year-old can lead to further studies on the bone marrow fat content using other mammals (although there were some studies already reported) as well. Except a few minor corrections, this manuscript can be published:

  1. Line 83, check the sentence once again and correct it accordingly.
  2. Line 85, remove extra period
  3. Conclusions section writing should be edited a bit more.
  4. Check spacing between words.

Author Response

Rec 2

The manuscript "Composition of fatty acids in bone marrow of red deer from 2 various ecosystems" is well written and thoroughly studied and investigated. Although the novelty of the manuscript is not high, the results of this manuscript in studying the composition of fatty acids in bone marrow of farmed vs. wild red deer fawns vs. 3-6-year-old can lead to further studies on the bone marrow fat content using other mammals (although there were some studies already reported) as well. Except a few minor corrections, this manuscript can be published:

  1. Line 83, check the sentence once again and correct it accordingly.
  2. Line 85, remove extra period
  3. Conclusions section writing should be edited a bit more.
  4. Check spacing between words.

Answ. Authors would like to thank the Reviewer for all valuable comments that increase the quality of this manuscript. All suggestions were included in the revised version of the manuscript. The conclusions have been rewritten to more detailed reflect the results of our research. 

Reviewer 3 Report

This study evaluated Composition of fatty acids in bone marrow of red deer from 2
various ecosystems. The topic is interesting and relevant, and major changes are needed to make this manuscript acceptable for publication in Molecules:

1.The abstract is very repetitive and confusing. Avoid repeating the same word several times in one sentence.

2.In the Introduction a better comprehension could be achieved with some connection between topics.

3.Line 89 - Why you did not choose the deer in the same age? You know, the age may affect the results.

4.Results - this part needs a big English language and grammar revision, or it will be difficult to follow and understand.

5.Line 124 - I think -80ºC is better than -20ºC.

6.Line 148 - add s blank between 260 and ºC.

7.Line 171 - ‘n=2’, the number of repetion is not enough.

8.I think Fig. 1 is not completed and it does not coutain the information of wild and farm deer sample.

Author Response

Rec 3

This study evaluated Composition of fatty acids in bone marrow of red deer from 2
various ecosystems. The topic is interesting and relevant, and major changes are needed to make this manuscript acceptable for publication in Molecules:

1.The abstract is very repetitive and confusing. Avoid repeating the same word several times in one sentence.

2.In the Introduction a better comprehension could be achieved with some connection between topics.

Answ. Authors would like to thank the Reviewer for all valuable comments that increase the quality of this manuscript. All suggestions were included in the revised version of the manuscript. The abstract, introduction and conclusions have been rewritten to more detailed reflect the results of our research.

3.Line 89 - Why you did not choose the deer in the same age? You know, the age may affect the results.

Answ. Thank you for your attention. Yes you are right, age can have an influence on the results obtained. Basically, it is not possible to obtain bone marrow samples from farmed does 6-7 month old  because does are intended for herd reconstruction and older too. Therefore, the animals were grouped in two groups: depending on the housing system, wild vs. farm and aged, wild  fawns vs. wild does. Moreover we harvested sampled wild elders does to see how changes occur with age.

4.Results - this part needs a big English language and grammar revision, or it will be difficult to follow and understand.

Answ. The paper has been subjected to linguistic corrections.

5.Line 124 - I think -80ºC is better than -20ºC.

Answ. It was improved. Please see line 156.

6.Line 148 - add s blank between 260 and ºC.

Answ. It was improved. Please see line 270.

7.Line 171 - ‘n=2’, the number of repetion is not enough.

Answ. We would like to inform that Statistical analysis section was redrafted and clarifying information was included.

8.I think Fig. 1 is not completed and it does not coutain the information of wild and farm deer sample.

Answ. The results were supplemented with a figure 2 reflecting more detailed analysis between the studied groups of animals.

Reviewer 4 Report

General

The paper shows many unclear points throughout the text and tables.

The research compared animals of different age, sex and physiological status also (pregnant, also?), thus the title is not appropriate. 

The paper does not show details on the date of slaughtering of the animals, and it seems that they were killed at different times ( and at different environmental conditions?). It is unclear why these groups were compared.

The body weight of the wild animals was estimated; why the same percentage for animals of different age? 

It seems inappropriate, for the results, a PCA analysis using estimated body weight.

Problems with the number of rows from page 9.

Main indications:

abstract: the level of significance is not given.

Some sentences are repeated in the text:

rows: 37-38, 49-51, 112-113 pg. 12, 104-107, 117-119. 

row 57, 81-85, 98-100, 176-177, 136 pag. 13: unclear

row 175: where is it in table 1?

Tables and figures: the groups are not well indicated (table 1); captions incomplete and unclear

row 156-167: they are important as captions for the tables.

Author Response

Rec 4

General

The paper shows many unclear points throughout the text and tables.

The research compared animals of different age, sex and physiological status also (pregnant, also?), thus the title is not appropriate. 

Answ. Authors would like to thank the Reviewer for all valuable comments that increase the quality of this manuscript. Basically, it is not possible to obtain bone marrow samples from farmed does 6-7 month old  because does are intended for herd reconstruction and older too. Therefore, the animals were grouped in two groups: depending on the housing system, wild vs. farm and aged, wild  fawns vs. wild does. Moreover we harvested sampled wild elders does to see how changes occur with age.

The paper does not show details on the date of slaughtering of the animals, and it seems that they were killed at different times ( and at different environmental conditions?). It is unclear why these groups were compared.

Answ. All animals were slaughter in November, both farm and wild, the differences in the time of obtaining the animals were not greater than 2 weeks. Hunters can hunting  seasons on the wild fawns red deer and does red deer from September 1 to January 15,  in accordance with the applicable Rules of Individual and Population Game Animal Selection in Poland (Polish Journal of Laws, Annex to Resolution No. 57/2005 of February 22, 2005).

The body weight of the wild animals was estimated; why the same percentage for animals of different age? 

Answ. In wild animals slaughter from the wild, it is not possible to measure body weight because the carcass has to be gutted. Therefore, body weight was estimated in accordance with generally accepted recommendations.

It seems inappropriate, for the results, a PCA analysis using estimated body weight.

Answ. Thank you very much for valuable comment. However, in fact, any analytical data is subject to measurement error, related to limitations in method accuracy or misestimation. Principal component analysis (PCA) is a technique used to emphasize variation and bring out strong patterns in a dataset, and is used to make data easy to explore and visualize. And this was the additional purpose of using this statistical analysis in the present study. Furthermore, the PCA analysis has been supplemented with a new figure taking into account the distribution of samples according to the group of animals evaluated.

Problems with the number of rows from page 9.

Main indications:

abstract: the level of significance is not given.

Answ. The abstract have been rewritten to more detailed reflect the results of our research.

Some sentences are repeated in the text:

rows: 37-38, 49-51, 112-113 pg. 12, 104-107, 117-119. 

row 57, 81-85, 98-100, 176-177, 136 pag. 13: unclear

Answ. The introduction and conclusions have been rewritten to more detailed reflect the results of our research. The paper has been subjected to linguistic corrections.

row 175: where is it in table 1?

Tables and figures: the groups are not well indicated (table 1); captions incomplete and unclear

row 156-167: they are important as captions for the tables.

Answ. Tables and figures has been improved. We hope that the results presented in this way will be clearer.

Round 2

Reviewer 1 Report

The inconsistencies are still remaining in the revised manuscript. It is unclear why the authors start describing the data presented in Table 3 instead from Table 2. Why the visualization description of differences between the red deer groups by score plots (Figure 2) are inserted in the text describing Figure 1?

I do not understand why the same data is twice repeated in all the tables (means and SD of all wild fawn variables in 4-5 and 7- 8 columns are the same) when the comparisons between farmed fawn x wild fawn and between wild fawn x wild does show appropriate p-values. This is an oversupply of the data.

Since the Table 3 does not fit on the page, it may need to be split again. Why not to think about separate tables for saturated and unsaturated fatty acids?

Why does Table 4 list all 17 components when only 3 of them are valid (eigenvalues are higher than 1)?

Author Response

Rev 1

Dear Reviewer,

The authors would like to warmly thank you for all comments and suggestions, especially the critical ones, aimed at improving the scientific value of the article and eliminating the most important errors. We greatly appreciate the opportunity that we have been given to further revise the manuscript. We believe that you will share the arguments submitted by authors and find this revision fully satisfactory.

The inconsistencies are still remaining in the revised manuscript. It is unclear why the authors start describing the data presented in Table 3 instead from Table 2.

and

Since the Table 3 does not fit on the page, it may need to be split again. Why not to think about separate tables for saturated and unsaturated fatty acids?

Answ. Thank you for these suggestions, the relevant amendments were introduced into the manuscript and the new layout of table was applied. Indeed, the results presented in this way are more readability.

Why the visualization description of differences between the red deer groups by score plots (Figure 2) are inserted in the text describing Figure 1?

Answ. Figure 2 is an interpretative supplement to Figure 1. Besides, it was additionally included in the paper at the request of one of the reviewers. One is a projection of the variables in the layout of the first two most important principal components, the other shows the distribution of cases for the same components. On the other hand, the relationships (correlations) occurring in this layout are constant, i.e. independent of their visualisation, thus data from both charts can be taken into account in the interpretation of the results.   

I do not understand why the same data is twice repeated in all the tables (means and SD of all wild fawn variables in 4-5 and 7- 8 columns are the same) when the comparisons between farmed fawn x wild fawn and between wild fawn x wild does show appropriate p-values. This is an oversupply of the data.

Answ. Thank you very much for this valuable suggestion, the relevant changes has been introduced into table.

Why does Table 4 list all 17 components when only 3 of them are valid (eigenvalues are higher than 1)?

Answ. It is true that only three are significant, although it is accepted to provide all values of identified components, irrespective of their values

Reviewer 3 Report

The paper can be published.

Author Response

Rev 3

Answ. Thank you kindly for your positive opinion.

Reviewer 4 Report

The paper has been improved and now it is more clearly presented. 

In my opinion the experimental design is not completely appropriate for the aim of the research. The comparison between young males and adult females may be affected by some different physiological aspects, not only the age. The title does not indicate the effect of age. 

For the aim of the study (row 76-78) the second effect is the age, but it is unclear , because adult does are cited. 

Furthermore, the body weight was estimated using a single value, referred both to young and adult animals. 

The abstract does not show the p level.

In the introduction, or in the text, indications on the body fat reserves of the different groups of deer could be useful. 

Row 135-139 vs 148-152.

Author Response

Rev 4

Dear Reviewer,

The authors would like to warmly thank you for all comments and suggestions, especially the critical ones, aimed at improving the scientific value of the article and eliminating the most important errors. We greatly appreciate the opportunity that we have been given to further revise the manuscript. We believe that you will share the arguments submitted by authors and find this revision fully satisfactory.

The paper has been improved and now it is more clearly presented.

Answ. Thank you kindly for your positive comment.

In my opinion the experimental design is not completely appropriate for the aim of the research. The comparison between young males and adult females may be affected by some different physiological aspects, not only the age. The title does not indicate the effect of age.

For the aim of the study (row 76-78) the second effect is the age, but it is unclear , because adult does are cited.

Answ. Thank you for this valuable suggestion. The animal category, which in animal science groups individuals of the same sex and similar age, was used as a variation factor. This factor was also included in the title of the paper and the aim of study.

Furthermore, the body weight was estimated using a single value, referred both to young and adult animals.

Answ. Thank you for this remark. However, due to the specific conditions and the possibility of taking standard post-mortem measurements as for animals killed in a slaughterhouse, the authors assumed an average dressing percentage of 67% for venison. Of course, the authors are aware of the consequences of such an estimation, treating the results obtained as preliminary for further in-depth analyses.   

The abstract does not show the p level.

Answ. The p levels were added for values in the abstract.

In the introduction, or in the text, indications on the body fat reserves of the different groups of deer could be useful.

Answ. Thank you for this valuable suggestion. We would like to inform, that the additional facts with relevant references (Dzięciołowski et al. [10]) were added in Introduction
